# Differential Cytotoxicity of Surface-Functionalized Silver Nanoparticles in Colorectal Cancer and Ex-Vivo Healthy Colonocyte Models

**DOI:** 10.3390/cancers17091475

**Published:** 2025-04-27

**Authors:** Marianna Barbalinardo, Emilia Benvenuti, Denis Gentili, Francesca Chiarini, Jessika Bertacchini, Luca Roncucci, Paola Sena

**Affiliations:** 1Consiglio Nazionale Delle Ricerche, Istituto per lo Studio dei Materiali Nanostrutturati (CNR-ISMN), Via P. Gobetti 101, 40129 Bologna, Italy; marianna.barbalinardo@cnr.it (M.B.); emilia.benvenuti@cnr.it (E.B.); 2Department of Biomedical, Metabolic and Neural Sciences, Section of Human Morphology, University of Modena and Reggio Emilia, Via del Pozzo 71, 41124 Modena, Italy; francesca.chiarini@unimore.it; 3Department of Surgery, Medicine Dentistry and Morphological Sciences with Interest in Transplant, University of Modena and Reggio Emilia, Via del Pozzo 71, 41124 Modena, Italy; jessika.bertacchini@unimore.it; 4Department of Medical and Surgical Sciences, University of Modena and Reggio Emilia, 41124 Modena, Italy; roncucci.luca@unimore.it

**Keywords:** colorectal cancer cells, human colonocyte, silver nanoparticles, ex-vivo, cytotoxicity

## Abstract

This study investigates the use of silver nanoparticles as a potential new treatment for colorectal cancer. Colorectal cancer is one of the most common cancers worldwide, and finding more effective treatments is essential. The researchers tested silver nanoparticles AgNPs with two different surface coatings to see how they affect cancer cells compared to healthy cells. One type of nanoparticles showed significant effects, reducing cancer cell growth and inducing cell death, while the other had minimal impact. These findings suggest that modifying the surface of nanoparticles could help target cancer cells more specifically, leading to treatments that are both more effective and have fewer side effects. This research could pave the way for new therapies for colorectal cancer and other types of cancer, ultimately improving patient outcomes and advancing cancer treatment strategies.

## 1. Introduction

In recent years, engineered nanomaterials have attracted significant attention due to their impact on oncology, enhancing tumor targeting, improving the delivery of therapeutic agents, and minimizing off-target effects [1]. Currently, the FDA has approved targeted therapies based on monoclonal antibodies (mAbs)—such as bevacizumab and cetuximab, which inhibit pathways crucial for tumor growth—angiogenesis through vascular endothelial growth factor (VEGF) inhibition, and the epidermal growth factor receptor (EGFR) [2]. Although effective when combined with chemotherapy, these therapies are hampered by issues of resistance and adverse effects, driving ongoing research into alternative nano-enabled treatments [3]. Nanoparticles (NPs) have attracted significant attention in biomedical research due to their potential to overcome the limitations of conventional treatments, such as poor bioavailability and low target specificity. These unique properties make NPs highly appealing for applications in cancer prevention, diagnosis, and therapy [4]. Among the various types of nanoparticles, silver nanoparticles (AgNPs) stand out for their distinctive characteristics, enabling their use in antimicrobial therapies, diagnostics, and drug delivery systems [5]. Recent studies have emphasized the importance of surface modification in optimizing the interactions of AgNPs with biological systems, highlighting it as a critical area of investigation [6]. Numerous studies have demonstrated that the choice of coating agent significantly influences selective cytotoxicity, with findings indicating that stronger binding between the metallic surface and the coating agent generally results in reduced toxicity [7,8]. For example, citrate-capped AgNPs (AgNPs-cit) promote protein adsorption, forming a ‘protein corona’ that enhances cellular uptake and induces dose-dependent cytotoxicity. In contrast, AgNPs coated with oligo (ethylene glycol)-alkanethiol (OEG), such as (11-mercaptoundecyl) hexa (ethylene glycol) (EG_6_OH), resist protein adsorption, significantly reducing both cellular uptake and cytotoxicity [9]. Modulating the density of EG_6_OH on AgNP surfaces represents a promising strategy to regulate protein corona formation, thereby influencing cellular uptake and associated cytotoxic effects [10]. Among the most common cancers globally, colorectal cancer certainly holds a significant position; it is the fourth most common cancer and the third leading cause of death [11]. The complex heterogeneity of CRC pathogenesis and the poor patient tolerance to current standard therapies are key factors contributing to the failure of pharmacological treatments and the rising mortality rate. Different subtypes of CRC cell lines exhibit a well-documented molecular heterogeneity, which leads to varied responses to therapeutic agents [12,13]. Despite notable research progress demonstrating how, through passive or active targeting, AgNPs selectively enter CRC cells via diffusion, phagocytosis, or endocytosis [14], there are still knowledge gaps regarding the cytotoxic effects of different types of functionalized nanoparticles coated with various ligands. Moreover, the impact of these nanoparticles on primary colonocyte cultures derived from normal mucosal tissues remains largely unexplored. Investigating whether non-tumoral cells exhibit a distinct response to these agents is crucial for a more comprehensive understanding of their therapeutic potential and safety. Herein, we explore the cytotoxic effects of two types of AgNPs, citrate- and EG_6_OH-coated, on CRC cell lines and healthy colonocytes derived from primary cultures of normal mucosal tissue. Using LoVo (microsatellite instability-high) and HT-29 cell lines alongside primary colonocyte cultures, the study reveals that surface functionalization critically influences nanoparticle interactions with biological systems. AgNPs-cit demonstrated significant cytotoxicity in LoVo cells, reducing viability and inducing morphological changes indicative of programmed cell death, particularly after 48 h of exposure. In contrast, AgNPs-EG_6_OH showed minimal toxicity. Interestingly, neither type of nanoparticle affected HT-29 cells or healthy colonocytes, underscoring the selective impact of AgNPs-cit on specific CRC subtypes. 

## 2. Materials and Methods

*Materials*. All chemicals were purchased from Merck and used without further purification, and aqueous solutions were prepared with deionized water (Milli-Q, Millipore, Burlington, MA, USA) with resistivity above 18 MΩ.

*Synthesis of citrate-coated AgNPs*. AgNPs-cit were prepared following the method reported elsewhere [15].

*Synthesis of EG_6_OH-coated AgNPs*. EG_6_OH-coated silver nanoparticles (AgNPs-EG_6_OH) were synthesized following previously established protocols [9,10]. Briefly, citrate-stabilized silver nanoparticles (AgNPs-cit) were mixed overnight at room temperature with a concentrated aqueous EG_6_OH solution (2 mM). Unbound ligands were removed by three cycles of centrifugation at 13,000× *g* for 30 min, followed by resuspension in a 2 mM aqueous sodium citrate solution.

*Characterization of NPs*. UV-vis spectra were recorded on a Jasco V-550 UV-Vis spectrophotometer. DLS and *ζ*-potential measurement were performed in phosphate buffer (1 mM, pH = 7) and KCl (1 mM) on a NanoBrook Omni Particle Size Analyzer (Brookhaven Instruments Corporation, Nashua, NH, USA).

*Human colon tissue collection and primary cell culture*. Biopsies of normal colorectal mucosa were collected from 12 healthy patients undergoing routine colonoscopies at the University Hospital of Modena, prompted by positive fecal occult blood tests and/or abdominal symptoms. All participants had no prior history of cancer or inflammatory bowel disease, and only those with one or more histologically confirmed adenomas identified during the colonoscopy were included in the study.

All participants were enrolled between 1 June 2021 and 1 February 2022. Exclusion criteria included patients under 18 years of age or with any type of malignancy. The study cohort consisted of six women and six men, with an average age of 60.8 years (M/F: 60.8/60.9). Approval for the study was obtained from the competent Ethics Committee (code no. 245/11) and the Local Health Agency of Modena, Italy. Each participant provided signed, informed consent. The study adhered to the Declaration of Helsinki Good Clinical Practice guidelines for medical research, and current regulations on the protection and processing of personal data (European Regulation No. 679/2016). For each patient, three samples of normal colonic mucosa were obtained using standard endoscopic biopsy forceps (Endo Jaw 2.8 mm, Olympus, Hamburg, Germany) and immediately immersed in 2 mL of IntestiCult Organoid Growth Medium (Human) (Stemcell Technologies, Vancouver, BC, Canada). To isolate crypt and glandular units, the surgical specimens were finely minced with a scalpel, and the resulting cell suspension was repeatedly pipetted to enhance cell dissociation. Cells (1 × 10^4^ per well) were seeded into 96-well plates, incubated with organoid growth medium (IntestiCult Organoid Differentiation Medium-Human; Stemcell Technologies), and cultured for 24 and 48 h at 37 °C in a 5% CO_2_ and 95% air environment.

*Human secondary cell culture*. Human colon cancer cells, HT-29, were cultured under standard conditions in DMEM medium supplemented with 10% (*v*/*v*) FBS, 2 mM L-glutamine, 100 U mL^−1^ penicillin, and 100 U mL^−1^ streptomycin in a humidified incubator set at 37 °C with 5% CO_2_. LoVo human colorectal cancer cells were maintained under standard conditions in RPMI 1640 medium (Roswell Park Memorial Institute, Buffalo, NY, USA) supplemented with 10% (*v*/*v*) fetal bovine serum (FBS), 2 mM L-glutamine, and antibiotics (100 U/mL penicillin and 100 U/mL streptomycin). Cultures were incubated at 37 °C in a humidified atmosphere containing 5% CO_2_. Cells were seeded into 96-well plates at a density of 5 × 10^4^ cells/mL and allowed to grow for 24 h prior to nanoparticle treatment. Control groups received only the vehicle solution.

*Treatments of human cell culture*. Cells collected from patient biopsies were treated with AgNPs-EG_6_OH and AgNPs-cit at a 20 µg/mL concentration for 24 and 48 h. Each treatment with both types of nanoparticles was conducted in triplicate for every patient. For the control, the cells were exposed to the vehicle solution.

*Cell proliferation assay* (*MTT*). After 24 h of treatment with AgNPs-EG_6_OH and AgNPs-cit, colonocyte morphology was observed under an inverted microscope (Olympus BX-51; Olympus Optical Co., Ltd., Tokyo, Japan). Cell proliferation of human colonocytes after 24 and 48 h of treatment with AgNPs-EG_6_OH and AgNPs-cit was evaluated using the MTT assay [3-(4,5-Dimethylthiazol-2-yl)-2,5-diphenyltetrazolium bromide] (Roche Diagnostics, Basel, Switzerland), following the manufacturer’s instructions. The optical density (OD) of each sample was recorded at 490 nm, and cell viability was determined using the formula: Cell viability (%) = (OD treated/OD control) × 100.

*Immunofluorescence by Optical and Confocal Microscopy*. Cells were cultured on two-well glass chamber slides for immunofluorescence analysis. Once a monolayer was established, cells were fixed with 4% paraformaldehyde in PBS for 10 min. This was followed by a 30 min blocking step using 3% BSA in PBS at room temperature. Subsequently, the cells were incubated with Rhodamine-conjugated phalloidin (ABCAM, Boston, MA, USA) at a 1:25 dilution in PBS containing 3% BSA for 1 h at room temperature. Phalloidin binds specifically to filamentous actin (F-actin), allowing visualization of actin filament organization. After washing with PBS, nuclei were counterstained with 1 µg/mL DAPI in water and the samples were mounted using an antifade medium composed of 0.21 M DABCO and 90% glycerol in 0.02 M Tris buffer (pH 8.0). Negative controls omitted primary antibody incubation. Confocal microscopy was performed using a Leica TCS SP2 AOBS confocal laser scanning microscope (Leica, Wetzlar, Germany) as previously described [16], and optical fluorescence imaging was conducted with a Nikon Eclipse 80i microscope (Nikon, Tokyo, Japan).

*Statistical analysis*. All data were statistically analyzed by using GraphPad Prism 8 software. Statistical comparisons were performed using unpaired *t*-test for two-group comparisons or one-way ANOVA followed by Tukey’s and Bonferroni post hoc tests for comparisons of three or more groups. Selectivity was assessed by comparing cell viability reductions in LoVo vs. HT-29 and primary colonocytes. Cell viability reduction was expressed as a percentage relative to the vehicle control. Data are presented as mean ± standard deviation (SD). A *p* value of less than 0.05 was considered statistically significant. To ensure statistical reliability, all experiments were performed in a minimum of three independent replicates, each with technical triplicates per condition.

## 3. Results

### 3.1. Synthesis and Functionalization of AgNPs

Citrate-coated silver nanoparticles (AgNPs-cit), with a hydrodynamic diameter of 19 ± 1 nm and a zeta potential of −44 ± 3 mV, were prepared and functionalized with EG_6_OH via ligand exchange (Figure 1a), as previously reported (see Experimental Section) [9,10]. As shown in Figure 1b, AgNPs-cit exhibited a characteristic surface plasmon resonance band centered at 400 nm. After ligand exchange with EG_6_OH, this band showed a slight redshift without broadening or shape variation, indicating that the aggregation state of the nanoparticles remains unaffected by functionalization [17]. Additionally, a slight increase in hydrodynamic diameter (22 ± 1 nm) and a drastic decrease in surface charge (−19 ± 3 mV) were observed, as expected [9,10].

### 3.2. AgNPs-Cit and AgNPs-EG_6_OH Exhibit Distinct Cytotoxic Effects on LoVo Cells

To assess whether the two types of nanoparticles—one coated with EG_6_OH and the other with citrate—had a cytotoxic effect on colorectal cancer cells, we performed a series of experiments in which the cells were incubated with a specific concentration of nanoparticles (20 µg/mL) for 24 and 48 h. This concentration of AgNPs, according to previously reported results, allows us to distinguish the role of surface coating in the interaction of AgNPs with cells, as under these conditions, AgNPs-cit are toxic, whereas AgNPs completely coated with EG_6_OH do not significantly affect cell viability [9,10]. Specifically, we utilized the following cell line: LoVo, a microsatellite instability-high (MSI-H) cell line harboring a KRAS mutation (G13D). The cytotoxic effect of treatment with the two types of nanoparticles was evaluated using the MTT colorimetric assay to determine the cell viability percentage relative to a control sample. As shown in the histogram in Figure 2, treatment of LoVo cells with AgNPs-cit for 24 h leads to a significant reduction in cell viability compared to the control. In contrast, treatment with AgNPs-EG_6_OH does not cause a significant decrease in cell viability under the same conditions. Moreover, when comparing the viability percentages of LoVo cells treated with the two types of nanoparticles for 24 h, a significant difference in the effects of AgNPs-cit and AgNPs-EG_6_OH is observed. After 48 h (Figure 2), the effect of treatment with AgNPs-cit on the number of viable cells becomes more pronounced and reaches greater statistical significance. On the other hand, the treatment with AgNPs-EG_6_OH does not exhibit any antiproliferative effects on cultured cells, even after 48 h. The difference in the number of viable cells between those treated with AgNPs-cit and those treated with AgNPs-EG_6_OH increases significantly. These results were also clearly observable using fluorescence confocal microscopy, which confirmed the differences in cell viability between treatments (Figure 2b–d).

### 3.3. AgNPs-Cit and AgNPs-EG_6_OH Do Not Exert a Negative Effect on the Cell Viability of HT-29 Cells 

The cytotoxic effect of both types of NPs was also evaluated on the HT29 cell line, an MSS cell line carrying critical mutations in BRAF, APC, and TP53. This makes it a valuable model for studying colorectal cancer that is not associated with microsatellite instability. The treatment was carried out under the same conditions used for the LoVo cells, but the results were completely different (Figure 3). Neither the 24 h nor the 48 h treatment with AgNPs-cit and AgNPs-EG_6_OH caused any significant changes in the number of viable cells compared to the control sample (Figure 3). These results were further corroborated through fluorescence optical microscopy, which provided clear visualization of the differences in cell viability between the treatments (Figure 3b–d).

### 3.4. AgNPs-Cit and AgNPs-EG_6_OH Do Not Show Any Antiproliferative Activity Against Human Colonocytes from Healthy Donors

Primary colonocyte cultures were obtained from the normal mucosa of 12 healthy patients, all of whom tested positive for the presence of at least one adenoma during endoscopic examination. Human colonocytes were treated for 24 and 48 h with the same concentration of AgNPs-cit and AgNPs-EG_6_OH used for the previously described colorectal cancer cell lines. The experimental results are summarized in the histogram in Figure 4, which clearly shows that treatment with the two different types of nanoparticles does not lead to a significant reduction in the cell proliferation rate compared to the untreated samples.

### 3.5. Morphological Changes Due to the Cytotoxic Effect of AgNPs-Cit on LoVo Cells Are Clearly Observable

The treatment with AgNPs-cit leads to significant cytotoxic effects on LoVo cells, as demonstrated by cell viability assays. These results are also clearly observable at the morphological level, as shown in Figure 5b, at 48 h after treatment with AgNPs-cit. LoVo cells undergo distinct morphological changes indicative of programmed cell death. The cytoplasm shrinks and becomes denser, while the cells lose their characteristic elongated or polygonal shape, adopting a more rounded and contracted appearance. Additionally, prominent membrane blebbing occurs, and a significant number of cells progressively detach from the monolayer. In contrast, no evident morphological changes were observed in HT-29 cells (Figure 5e) or human colonocytes treated with either of the nanoparticles.

### 3.6. Internalization of Both AgNPs-Cit and AgNPs-EG_6_OH in Colonocytes Assessed Using Confocal Microscopy Techniques

Experiments have been designed to evaluate the potential internalization of both AgNPs-cit and AgNPs-EG_6_OH by colonocytes using confocal microscopy techniques, incorporating fluorochromes conjugated to phalloidin. The goal was to identify any dark regions within the actin cytoskeletal structure, which could suggest fluorescence quenching caused by the presence of metal nanoparticles. As shown in Figure 6, distinct dark areas are clearly visible within the cells, evenly distributed throughout the cytoplasm and varying in size, both in cells treated with AgNPs-cit and those treated with AgNPs-EG_6_OH when compared to the control samples. In contrast, the control samples show phalloidin staining as bright, intense areas, with well-distributed and organized actin filaments within the cell. These black spots are detectable as early as 24 h post-treatment and are more widely distributed within the cellular volume after 48 h of treatment.

## 4. Discussion

The results highlight the critical role of surface functionalization in modulating the biological effects of AgNPs. Citrate-coated AgNPs (AgNPs-cit) were successfully synthesized and, as highlighted by the slight redshift in the surface plasmon resonance (SPR) band, functionalized to achieve EG_6_OH-coated AgNPs (AgNPs-EG_6_OH) without inducing aggregation. This aspect is crucial because colloidal stability and lack of aggregation directly influence biological behavior and cytotoxic effectiveness [18]. 

Interestingly, AgNPs-cit exhibited significant cytotoxicity in LoVo cells, inducing apoptosis, as evidenced by altered morphology, membrane blebbing, and detachment from the substrate. The morphological changes observed in LoVo cells, prominently displayed in Figure 5, are characteristic features of caspase-dependent apoptosis [19]. This effect was found to be time dependent. In contrast, EG_6_OH-functionalized AgNPs did not cause any significant reduction in cell viability, suggesting that the coating influences the interaction mechanism between nanoparticles and cells. This disparity underscores how surface chemistry influences cellular interactions, potentially through mechanisms such as protein corona formation, oxidative stress generation, or differential cellular uptake [20]. AgNPs-cit has been reported to be rapidly coated by serum proteins, forming the so-called protein corona, which favors their cellular uptake and cytotoxic effects on NIH-3T3 cells [9]. In contrast, the EG_6_OH coating is known to reduce non-specific protein adsorption (thereby decreasing protein corona complexity) on the surface of nanoparticles and, similar to what has been reported for PEGylated nanoparticles [21,22], minimizes interactions with cellular receptors, thereby reducing the internalization of AgNPs and consequently their cytotoxicity [10]. Similarly, we can infer that the uptake of AgNPs-cit in LoVo cells is promoted by the formation of the protein corona. This heightened uptake contributes to the greater intracellular accumulation of AgNPs-cit in LoVo cells, amplifying their cytotoxic effects [23]. This is consistent with studies showing that citrate-coated nanoparticles have a higher affinity for membrane receptors involved in phagocytosis and endocytosis [24]. Interestingly, treatment with both types of nanoparticles showed no adverse effects on the proliferation of HT-29 cells. This discrepancy suggests that intrinsic differences in metabolic pathways, membrane characteristics, or nanoparticle internalization mechanisms could underlie the observed effects. 

LoVo and HT-29 cells represent distinct models of colorectal cancer [25,26], with LoVo cells characterized by microsatellite instability (MSI-H) and HT-29 cells displaying microsatellite stability (MSS) [27,28]. These differences are associated with distinct molecular profiles and cellular behaviors; MSI-H cells, like LoVo, often exhibit altered metabolic states compared to MSS cells [29]. This may influence their susceptibility to oxidative stress induced by AgNPs, as reactive oxygen species (ROS) detoxification pathways could differ significantly. Indeed, studies have shown that in LoVo cells, silver nanoparticles initiate a signaling cascade that results in the generation of ROS, leading to mitochondrial dysfunction in a size- and dose-dependent manner [30]. The increased ROS levels also contribute to cell cycle arrest and subsequently induce apoptosis. Furthermore, excessive ROS induces endoplasmic reticulum (ER) stress and reduces the mitochondrial membrane potential (MMP), thereby activating intrinsic pathways that drive apoptosis in tumor cells [31].

Moreover, variations in the composition of membrane lipids and proteins may affect the electrostatic interactions between the negatively charged citrate-coated AgNPs and the cell surface. For example, LoVo cells might have membrane regions more prone to destabilization or nanoparticle adsorption. On the other hand, differences in endocytic pathways or receptor expression could alter the uptake of AgNPs, affecting their intracellular accumulation and toxicity [32]. LoVo cells may internalize nanoparticles more effectively, leading to higher cytotoxic effects compared to HT-29 cells. 

HT-29 cells carry critical mutations in BRAF, APC, and TP53, which may further modulate their response to AgNP-induced stress [33]. The BRAF mutation (commonly V600E in HT-29 cells) activates the MAPK pathway, driving cell proliferation and potentially increasing resistance to oxidative stress [34,35]. This could limit the cytotoxic impact of AgNPs. 

APC mutations play a pivotal role in Wnt signaling dysregulation, which might affect endocytosis and membrane dynamics, potentially reducing nanoparticle uptake [36]. 

Mutations in TP53, a key regulator of the cellular stress response, could impair apoptosis induction. HT-29 cells with dysfunctional p53 may exhibit a dampened response to oxidative stress and reduced susceptibility to AgNP-induced apoptosis [37]. 

The higher susceptibility of LoVo cells to oxidative stress induced by AgNPs may be in line with the higher sensitivity of MSI-H gastrointestinal cancer cells to immune checkpoint inhibitors [38]. 

The broader implications of our observation underscore the importance of understanding how genetic and phenotypic differences influence nanoparticle-cell interactions. While LoVo cells are more susceptible to AgNP-induced cytotoxicity, HT-29 cells appear to be resistant. 

Another important consideration is that the results of this study provide valuable insights into the safety profile and selective cytotoxicity of citrate-coated silver nanoparticles and EG_6_OH-functionalized silver nanoparticles, particularly regarding their interactions with both tumor and healthy cells. 

The observation that both types of nanoparticles do not exhibit antiproliferative effects on healthy human colonocytes suggests that at the tested concentrations, AgNPs-cit and AgNPs-EG_6_OH are biocompatible and safe for normal, non-cancerous cells. This is a crucial finding, as one of the primary challenges in nanomedicine is ensuring that nanoparticles selectively target and affect only diseased cells, minimizing toxicity to healthy tissues. 

The absence of cytotoxicity toward colonocytes may be attributed to the lack of specific interaction between these nanoparticles and the normal cells. As mentioned above, tumor cells often have altered membrane compositions, enhanced endocytic uptake, and altered intracellular pathways that may make them more susceptible to nanoparticle-induced damage. In contrast, healthy cells may be more effective at shielding themselves from nanoparticle uptake or may possess more robust repair mechanisms to counteract oxidative stress induced by nanoparticles. Healthy cells may possess better antioxidant defense systems, such as higher expression of antioxidant enzymes, which could neutralize the ROS generated by nanoparticles, reducing the likelihood of cellular damage. Our results are consistent with those observed in several studies conducted, for example, on non-cancerous liver cells compared to liver cancer cell lines or on different types of tissues treated with nanoparticles coated with natural resin or pharmacologically active natural substances [39,40]. We aim to emphasize that the selection of LoVo (MSI-H) and HT-29 (MSS) cell lines, along with primary colonocytes, was based on their distinct molecular and biological characteristics, providing a robust model for assessing the selective cytotoxicity of AgNPs. The use of these three models allows for a well-rounded investigation into the impact of AgNPs, ensuring that findings are not limited to a single CRC subtype but instead provide insights applicable to a broader spectrum of colorectal cancer cases. Furthermore, this approach aligns with precision oncology efforts, where treatment strategies are tailored based on molecular subtypes of cancer. Moreover, this finding is significant for the potential therapeutic applications of differentially functionalized AgNPs, as it suggests that these nanoparticles can offer selectivity toward cancer cells while sparing normal, healthy tissues. 

The internalization of AgNPs-cit and AgNPs-EG_6_OH by cells, confirmed by the observation of dark areas in the cytoplasm via confocal microscopy, indicates that these nanoparticles are effectively taken up by cells, regardless of their surface coating, in both colorectal cancer and healthy cells. The dark areas observed are likely a result of nanoparticle accumulation within the cell, which can interfere with cellular processes. 

Despite the similar uptake, differences in nanoparticle distribution and cytotoxic efficacy between the two nanoparticle types could be due to post-internalization mechanisms. AgNPs-cit interact more efficiently with the negatively charged cell membrane due to their surface charge, promoting endocytic uptake and intracellular accumulation. For these reasons, as previously explained, AgNPs-cit may induce more pronounced oxidative stress upon internalization due to the enhanced interaction between the citrate coating and cellular membranes, while EG_6_OH-coated nanoparticles may induce a more controlled release of ROS or show limited interaction with organelles due to the steric hindrance provided by the PEG coating. Additionally, the internalized nanoparticles could potentially interfere with organelles, such as mitochondria, leading to disrupted cellular functions and apoptosis in tumor cells but with minimal impact on normal cells. 

## 5. Conclusions

This study emphasizes the crucial role of nanoparticle surface chemistry in determining their biological effects. Citrate-coated nanoparticles are more likely to induce cytotoxicity through enhanced membrane interactions and oxidative stress, while EG_6_OH-coated nanoparticles exhibit reduced cytotoxicity due to steric hindrance and less protein corona formation. Tailoring nanoparticle surface chemistry based on specific therapeutic or diagnostic needs is essential for optimizing cancer treatments. This study also suggests the importance of understanding how specific genetic mutations and cellular properties may influence nanoparticle endocytosis, reactive oxygen species (ROS) production, and apoptosis. Such mechanistic insights could lead to more effective and selective nanoparticle-based therapies for colorectal cancer and other malignancies. Particularly, AgNPs-cit show promise for targeting microsatellite instability-high (MSI-H) cancer cell lines, offering a potential strategy for selective cancer treatment. The results underline how surface coating significantly affects the biological behavior of nanoparticles and the need for targeted designs based on cancer type. Further evaluations in additional cancer cell lines and in vivo models are needed to confirm the findings. Additionally, EG_6_OH coating may contribute to improved tumor-specific enrichment while reducing systemic side effects. This differential response is fundamental in nanomedicine, where achieving tumor-specific targeting and minimizing harm to healthy cells are key to successful therapeutic outcomes. In conclusion, AgNPs demonstrate significant potential as selective therapeutic agents for cancer treatment. The differential effects between the two nanoparticle types underscore the importance of surface functionalization in modulating their cellular interactions and cytotoxicity. Future studies should focus on understanding the mechanisms behind selective internalization and cytotoxicity, as well as testing these nanoparticles in vivo to confirm their safety and therapeutic efficacy.

## Figures and Tables

**Figure 1 cancers-17-01475-f001:**
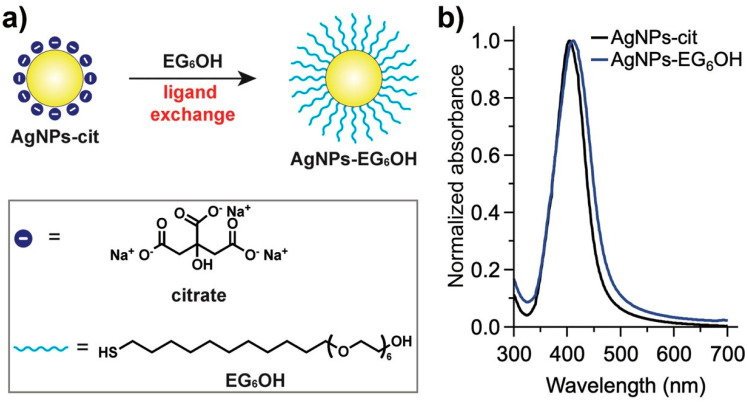
(**a**) Schematic representation of the ligand exchange process from AgNPs-cit to EG_6_OH-functionalized silver nanoparticles (AgNPs-EG_6_OH). The inset shows the chemical structures of citrate and EG_6_OH. (**b**) UV-vis absorbance spectra of AgNPs-cit (black line) and AgNPs-EG_6_OH (blue line).

**Figure 2 cancers-17-01475-f002:**
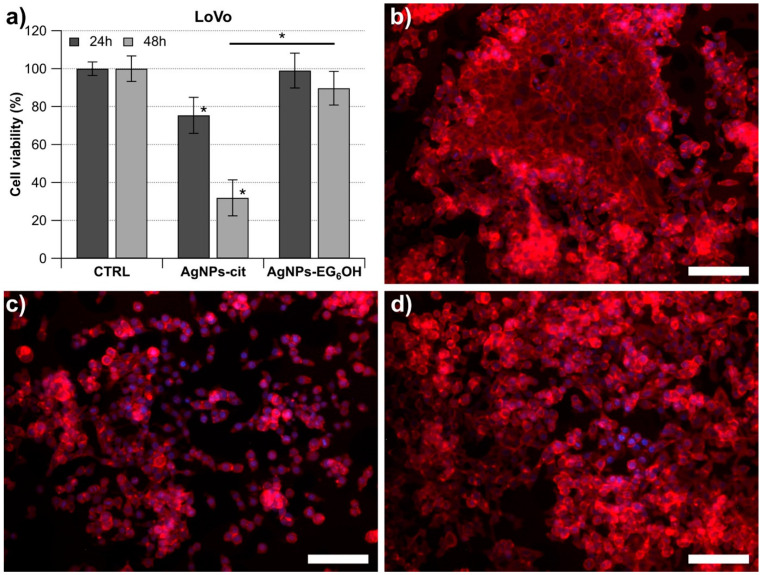
Cytotoxic effects of AgNPs-cit and AgNPs-EG_6_OH on LoVo cells. (**a**) LoVo cells were incubated with 20 µg/mL of AgNPs-cit or AgNPs-EG_6_OH for 24 and 48 h. Neither AgNPs-cit nor AgNPs-EG_6_OH caused any significant changes in cell viability compared to controls both at 24 and at 48 h. Asterisks indicate that the data are statistically different, with *p* < 0.05. Fluorescence microscopy images further confirmed the differences in cell viability between treatments. LoVo after 48 h: (**b**) control and treatment with (**c**) AgNPs-cit and (**d**) AgNPs-EG_6_OH (scale bar 100 µm).

**Figure 3 cancers-17-01475-f003:**
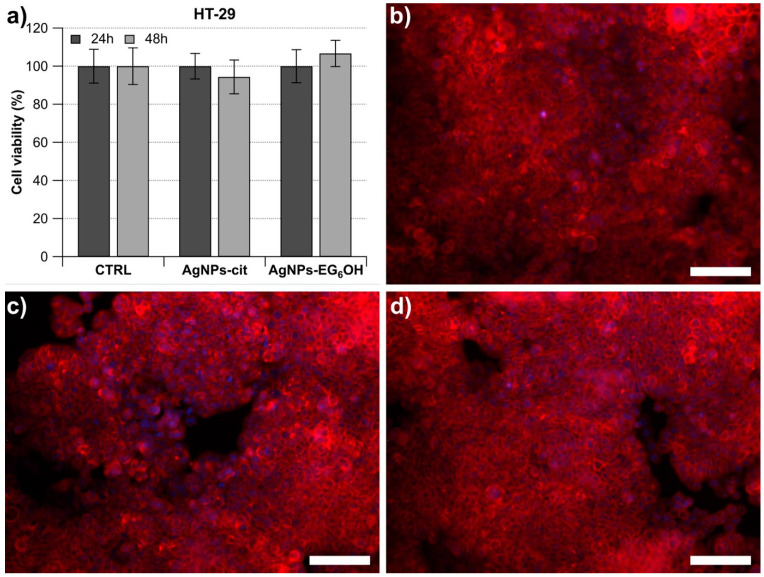
Effect of AgNPs-cit and AgNPs-EG_6_OH on HT-29 cell viability. (**a**) HT-29 cells were treated with 20 µg/mL of AgNPs-cit or AgNPs-EG_6_OH for 24 and 48 h. Neither AgNPs-cit nor AgNPs-EG_6_OH caused any significant changes in cell viability compared to controls. Fluorescence microscopy images confirm the absence of significant antiproliferative effects in both treatments. HT29 after 48 h: (**b**) control and treatments with (**c**) AgNPs-cit and (**d**) AgNPs-EG_6_OH (scale bar 100 µm).

**Figure 4 cancers-17-01475-f004:**
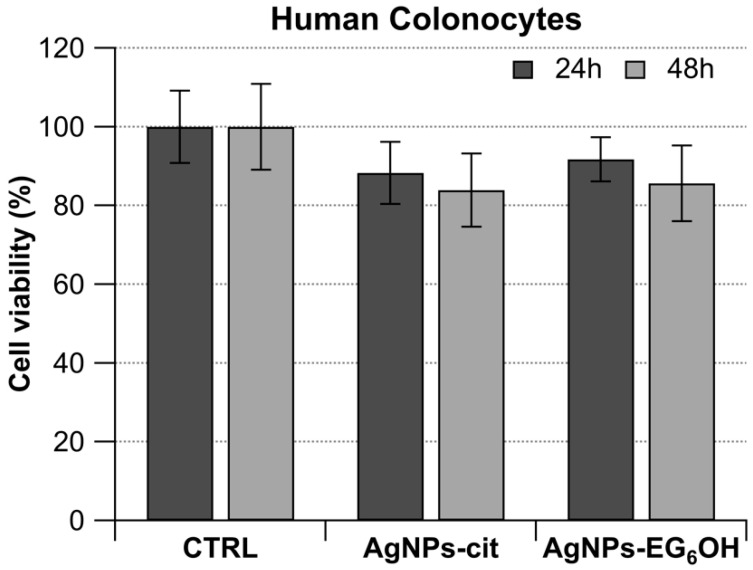
Histogram showing the effect of AgNPs-cit and AgNPs-EG_6_OH on primary colonocyte cultures. Colonocytes from 12 healthy patients, all with at least one adenoma, were treated with 20 µg/mL of AgNPs-cit or AgNPs-EG_6_OH for 24 and 48 h. Neither nanoparticle type caused a significant reduction in proliferation of cells compared to controls.

**Figure 5 cancers-17-01475-f005:**
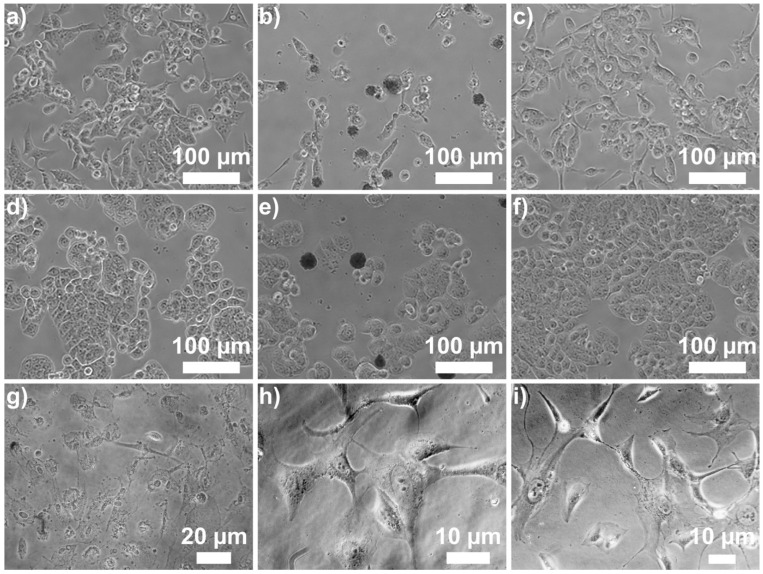
Optical micrographs of LoVo and HT-29 cells and human colonocytes after 48 h. LoVo: (**a**) control; treated with (**b**) AgNPs-cit and (**c**) AgNPs-EG_6_OH. HT-29: (**d**) control; treated with (**e**) AgNPs-cit and (**f**) AgNPs-EG_6_OH. Colonocytes: (**g**) control; treated with (**h**) AgNPs-cit and (**i**) AgNPs-EG_6_OH. Morphological changes in LoVo cells after AgNPs-cit treatment are evident at 48 h; LoVo cells exhibit signs of programmed cell death, including cytoplasmic shrinkage, cell rounding, and membrane blebbing. In contrast, no morphological alterations are observed in HT-29 cells or human colonocytes treated with AgNPs-cit or AgNPs-EG_6_OH.

**Figure 6 cancers-17-01475-f006:**
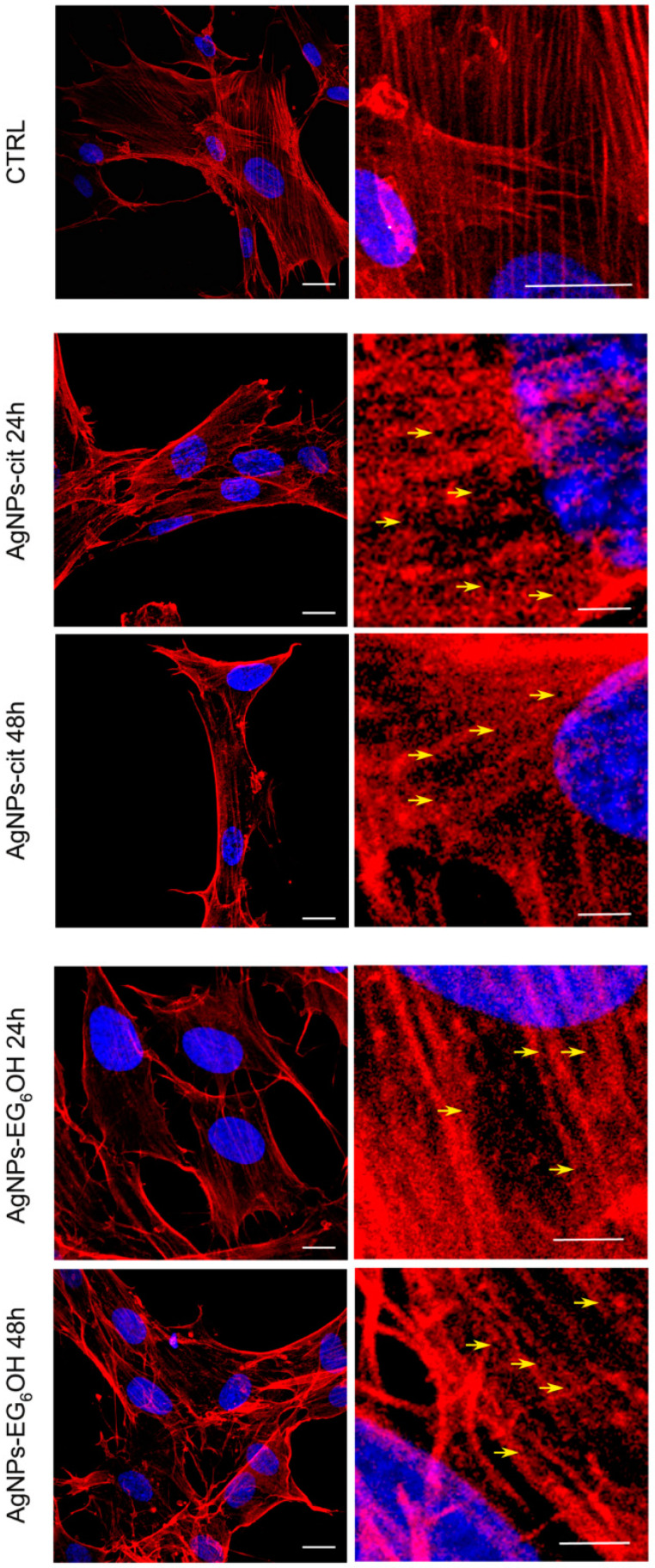
Internalization of AgNPs-cit and AgNPs-EG_6_OH in colonocytes assessed by confocal microscopy. Colonocytes were treated with AgNPs-cit or AgNPs-EG_6_OH, and fluorochromes conjugated to phalloidin were used to stain the actin cytoskeleton. Dark areas, indicating fluorescence quenching due to the presence of metal nanoparticles, are clearly visible in the cytoplasm of treated cells. These dark spots are evenly distributed and vary in size in both AgNPs-cit and AgNPs-EG_6_OH-treated cells, in contrast to the control samples, where phalloidin staining appears bright and intense, reflecting well-distributed and organized actin filaments. The dark spots appear as early as 24 h post-treatment and become more prominent and widespread after 48 h. Scale bars: in CTRL left column 10 µm and right column 10 µm; in AgNPs-cit 24/48 h and AgNPs-EG_6_OH 24/48 h left column 10 µm and column 1 µm.

## Data Availability

The data presented in this study are available upon request from the corresponding author. The data are not publicly available due to ethical restrictions.

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
