# Peer review of "Differential Cytotoxicity of Surface-Functionalized Silver Nanoparticles in Colorectal Cancer and Ex-Vivo Healthy Colonocyte Models"

_cancers, 2025, doi:10.3390/cancers17091475_

Round 1

Reviewer 1 Report (Previous Reviewer 1)

Comments and Suggestions for Authors

The authors have submitted all the changes asked by this review. 

Now the paper is ready to go. 

Author Response

We thank the reviewer for the positive assessment and for their valuable comments, which helped us improve the manuscript

Reviewer 2 Report (Previous Reviewer 2)

Comments and Suggestions for Authors

The authors have given an excellent reponse to my concerning points. Now, I think it could be accepted by this journal.

Author Response

We thank the reviewer for the positive assessment and for their valuable comments, which helped us improve the manuscript

Reviewer 3 Report (Previous Reviewer 3)

Comments and Suggestions for Authors

The author's response did not fundamentally address the existing problems. The author did not use more methods to characterize the nanoparticles. The manuscript did not add more experimental concentrations to test its cytotoxicity, and then calculate the IC50. In short, this article is not suitable for publication at this time.

Author Response

We respectfully disagree with the reviewer’s conclusion. In the revised manuscript, we have provided additional explanations and clarifications regarding the nanoparticle characterization and cytotoxicity assessment. Actually, we included additional characterization methods such as DLS and zeta potential measurements. We decided not to include further analyses because the nanoparticles have already been extensively characterized in previous studies, which we have referenced accordingly. While we acknowledge that further experiments (such as a full IC50 calculation) could expand the scope of the study, we believe that the data already included are scientifically sound, coherent, and sufficient to support the conclusions drawn. At this stage, we do not plan to perform additional experimental work, as we consider the study complete and aligned with the aims and scope of the journal.

This manuscript is a resubmission of an earlier submission. The following is a list of the peer review reports and author responses from that submission.

Round 1

Reviewer 1 Report

Comments and Suggestions for Authors

This paper presents an investigation into the use of silver nanoparticles as a potential new treatment for colorectal cancer. While the study is promising, I have several questions and suggestions that could enhance the final version.

  1. Please clarify the rationale for selecting LoVo and HT-29 cells, as well as primary colonocytes, in the study. Providing a more detailed explanation will strengthen the justification for these models, in the final version.
  2. The statistical analysis of AgNPs-cit cytotoxicity on microsatellite instability-high CRC cells should be more clearly described. How was selectivity assessed, and what statistical methods were used? Please include this information in the final version.
  3. While the paper presents evidence of a significant difference in cytotoxicity between AgNPs-cit and AgNPs-EG6OH, additional arguments would help reinforce this conclusion. Could the authors further elaborate on these differences?
  4. A mechanistic explanation for the selective cytotoxicity of AgNPs-cit would enhance the discussion. If available, please provide further details or cite a reference other than [reference 17] to support this claim.
  5. Is there a discussion on the potential pathways involved in AgNP-induced programmed cell death? If so, please ensure this is clearly presented.
  6. The methodology section should better explain how the authors accounted for differences in toxicity between AgNPs-cit and AgNPs-EG6OH. Providing more details would improve clarity.
  7. Finally, how does this study compare with previous research on AgNP cytotoxicity in CRC cells? A discussion on this aspect would help contextualize the findings.

I appreciate the authors’ efforts in this study and look forward to seeing these clarifications in the final version.

Reply to Reviewer 1’s comments 

We sincerely appreciate the time and effort you have dedicated to reviewing our manuscript. Your insightful comments and constructive suggestions have been invaluable in improving the clarity and scientific rigor of our work. We have carefully addressed each of your points and revised the manuscript accordingly to enhance its quality. Below, we provide a detailed response to each comment, outlining the modifications made and clarifying key aspects of our study. 

  1. Please clarify the rationale for selecting LoVo and HT-29 cells, as well as primary colonocytes, in the study. Providing a more detailed explanation will strengthen the justification for these models, in the final version. 

Reply: 

The selection of LoVo (MSI-H) and HT-29 (MSS) cell lines, along with primary colonocytes, was based on their distinct molecular and biological characteristics, which provide a comprehensive model for evaluating the selective cytotoxicity of AgNPs. 

LoVo cells (MSI-H, KRAS-mutant): These cells represent colorectal cancer with microsatellite instability, a subtype that is particularly relevant due to its distinct response to oxidative stress and targeted therapies. MSI-H tumors often exhibit defects in DNA mismatch repair, leading to increased mutation rates and unique vulnerabilities to ROS-mediated cytotoxicity. The presence of the KRAS G13D mutation further influences signaling pathways related to cell survival and apoptosis, making LoVo cells a crucial model for studying nanoparticle-induced cytotoxic effects. Patients with MSI-H tumors generally have a better prognosis compared to MSS cases, particularly in the early stages of the disease [1]. This improved prognosis is attributed to the high mutational burden of MSI-H tumors, which enhances immune system recognition and response [2]. However, MSI-H tumors are also known to be more responsive to immunotherapy, such as immune checkpoint inhibitors, highlighting the potential for combining nanoparticle-based therapies with existing treatments [3]. 

HT-29 cells (MSS, BRAF/APC/TP53-mutant): These cells model microsatellite-stable colorectal cancer, which is more common in clinical settings. The combination of BRAF, APC, and TP53 mutations affects cell proliferation, apoptosis resistance, and endocytic uptake of nanoparticles. Given that HT-29 cells showed resistance to AgNPs-cit treatment, their inclusion highlights the differential response based on genetic background, supporting the hypothesis that surface functionalization plays a key role in AgNP-mediated cytotoxicity. MSS tumors generally have a poorer prognosis compared to MSI-H cases as reported above, particularly when associated with BRAF mutations, which correlate with aggressive tumor behavior and lower survival rates [4]. 

Primary colonocytes (healthy donor-derived): These cells were included to evaluate the biocompatibility of AgNPs and assess whether their effects are selective for cancerous cells. Given that neither AgNPs-cit nor AgNPs-EG6OH induced significant cytotoxicity in primary colonocytes, the study provides evidence that AgNPs-cit selectively targets cancer cells without affecting normal colon tissue. This aspect is critical for potential therapeutic applications, as minimizing off-target effects is a key consideration in nanoparticle-based treatments. 

The use of these three models allows for a well-rounded investigation into the impact of AgNPs, ensuring that findings are not limited to a single CRC subtype but instead provide insights applicable to a broader spectrum of colorectal cancer cases. Furthermore, this approach aligns with precision oncology efforts, where treatment strategies are tailored based on molecular subtypes of cancer. 

  1. Xu Y, Liu K, Li C, Li M, Zhou X, Sun M, Zhang L, Wang S, Liu F, Xu Y. Microsatellite instability in mismatch repair proficient colorectal cancer: clinical features and underlying molecular mechanisms. EBioMedicine. 2024 May;103:105142. doi: 10.1016/j.ebiom.2024.105142. Epub 2024 Apr 30. PMID: 38691939; PMCID: PMC11070601. 
  1. Llosa NJ, Luber B, Tam AJ, Smith KN, Siegel N, Awan AH, Fan H, Oke T, Zhang J, Domingue J, Engle EL, Roberts CA, Bartlett BR, Aulakh LK, Thompson ED, Taube JM, Durham JN, Sears CL, Le DT, Diaz LA, Pardoll DM, Wang H, Anders RA, Housseau F. Intratumoral Adaptive Immunosuppression and Type 17 Immunity in Mismatch Repair Proficient Colorectal Tumors. Clin Cancer Res. 2019 Sep 1;25(17):5250-5259. doi: 10.1158/1078-0432.CCR-19-0114. Epub 2019 May 6. PMID: 31061070; PMCID: PMC6726531. 
  1. Marabelle A, Le DT, Ascierto PA, Di Giacomo AM, De Jesus-Acosta A, Delord JP, Geva R, Gottfried M, Penel N, Hansen AR, Piha-Paul SA, Doi T, Gao B, Chung HC, Lopez-Martin J, Bang YJ, Frommer RS, Shah M, Ghori R, Joe AK, Pruitt SK, Diaz LA Jr. Efficacy of Pembrolizumab in Patients With Noncolorectal High Microsatellite Instability/Mismatch Repair-Deficient Cancer: Results From the Phase II KEYNOTE-158 Study. J Clin Oncol. 2020 Jan 1;38(1):1-10. doi: 10.1200/JCO.19.02105. Epub 2019 Nov 4. PMID: 31682550; PMCID: PMC8184060.  
  1. Taieb J, Sinicrope FA, Pederson L, Lonardi S, Alberts SR, George TJ, Yothers G, Van Cutsem E, Saltz L, Ogino S, Kerr R, Yoshino T, Goldberg RM, André T, Laurent-Puig P, Shi Q. Different prognostic values of KRAS exon 2 submutations and BRAF V600E mutation in microsatellite stable (MSS) and unstable (MSI) stage III colon cancer: an ACCENT/IDEA pooled analysis of seven trials. Ann Oncol. 2023 Nov;34(11):1025-1034. doi: 10.1016/j.annonc.2023.08.006. Epub 2023 Aug 23. PMID: 37619846; PMCID: PMC10938565. 

Accordingly with the referee’s comment and to further clarify this point to the readers, we have integrated the discussion with additional sentences: We aim to emphasize that the selection of LoVo (MSI-H) and HT-29 (MSS) cell lines, along with primary colonocytes, was based on their distinct molecular and biological characteristics, providing a robust model for assessing the selective cytotoxicity of AgNPs. The use of these three models allows for a well-rounded investigation into the impact of AgNPs, ensuring that findings are not limited to a single CRC subtype but instead provide insights applicable to a broader spectrum of colorectal cancer cases. Furthermore, this approach aligns with precision oncology efforts, where treatment strategies are tailored based on molecular subtypes of cancer. 

  1. The statistical analysis of AgNPs-cit cytotoxicity on microsatellite instability-high CRC cells should be more clearly described. How was selectivity assessed, and what statistical methods were used? Please include this information in the final version. 

Reply: 

Statistical comparisons were performed using one-way ANOVA followed by Tukey's post hoc test for multiple group comparisons, ensuring robust statistical validation of differences between treated and control groups. For two-group comparisons, an unpaired t-test was used to determine significant differences in cytotoxicity between specific cell lines. Selectivity was assessed by comparing cell viability reductions in LoVo vs. HT-29 and primary colonocytes. The percentage reduction in cell viability was calculated relative to vehicle solution controls, and differences were considered statistically significant if p-values were below 0.05. Additionally, effect sizes were considered to determine the magnitude of cytotoxic differences between MSI-H and MSS cell lines, supporting the hypothesis that AgNPs-cit selectively affects MSI-H CRC cells. To determine the selectivity of AgNPs-cit for MSI-H colorectal cancer cells, statistical analyses were performed using multiple approaches to ensure robust data interpretation. 

Accordingly, the methodology section will be revised to explicitly describe these statistical methods in greater detail, ensuring transparency in the evaluation of AgNPs-cit selectivity for MSI-H CRC cells. 

  1. While the paper presents evidence of a significant difference in cytotoxicity between AgNPs-cit and AgNPs-EG6OH, additional arguments would help reinforce this conclusion. Could the authors further elaborate on these differences? 

Reply: 

AgNPs-cit exhibit higher cytotoxicity due to increased cellular uptake facilitated by protein corona formation, leading to enhanced oxidative stress. The presence of the citrate coating enhances the adsorption of serum proteins, forming a complex protein corona that facilitates receptor-mediated endocytosis, a key factor in cytotoxicity differences. Confocal microscopy analysis confirmed a higher degree of internalization of AgNPs-cit compared to AgNPs-EG6OH, further supporting their increased cytotoxic potential. AgNPs-EG6OH, with its sterically hindered surface, reduces protein adsorption, leading to lower uptake and minimal cytotoxicity. The EG6OH functionalization provides a hydrophilic and anti-fouling surface that inhibits protein corona formation, decreasing non-specific cellular interactions and subsequent internalization. AgNPs-cit could significantly increase ROS production in LoVo cells, while AgNPs-EG6OH might not induce measurable oxidative damage. This differential behavior is supported by previous studies demonstrating the role of surface functionalization in modulating Ag-nanoparticle interactions with biological systems [1].  

  1. Akter M, Sikder MT, Rahman MM, Ullah AKMA, Hossain KFB, Banik S, Hosokawa T, Saito T, Kurasaki M. A systematic review on silver nanoparticles-induced cytotoxicity: Physicochemical properties and perspectives. J Adv Res. 2017 Nov 2;9:1-16. doi: 10.1016/j.jare.2017.10.008. PMID: 30046482; PMCID: PMC6057238. 

To further strengthen this point, in the conclusion section, we have added the following sentences: AgNPs-cit interact more efficiently with the negatively charged cell membrane due to their surface charge, promoting endocytic uptake and intracellular accumulation. 

  1. A mechanistic explanation for the selective cytotoxicity of AgNPs-cit would enhance the discussion. If available, please provide further details or cite a reference other than [reference 17] to support this claim. 

Reply: 

Based on our previous studies and relevant literature, both cited in the article, we consider the following to be the most probable mechanistic aspect: AgNPs-cit rapidly adsorb serum proteins, forming a protein corona, which facilitates cellular recognition and internalization via endocytosis and macropinocytosis. This is consistent with studies showing that citrate-coated nanoparticles have a higher affinity for membrane receptors involved in phagocytosis and endocytosis [1]. MSI-H colorectal cancer (CRC) cells, such as LoVo, often exhibit increased macropinocytosis activity compared to microsatellite-stable (MSS) CRC cells like HT-29 [2]. This heightened uptake contributes to the greater intracellular accumulation of AgNPs-cit in LoVo cells, amplifying their cytotoxic effects. 

For completeness in our response, we present other possible mechanisms that we have not investigated in depth, but which are plausible on the basis of the literature: 

Induction of Oxidative Stress and ROS-Mediated Cytotoxicity: AgNPs-cit promote reactive oxygen species (ROS) generation once internalized, leading to mitochondrial damage, DNA fragmentation, and apoptosis [3]. MSI-H cells are characterized by defects in DNA mismatch repair (MMR), which impair their ability to efficiently respond to oxidative damage [4]. This may explain their increased susceptibility to AgNPs-cit-induced stress. The lack of cytotoxicity observed in HT-29 cells (MSS subtype) suggests a stronger intrinsic defense against oxidative stress, possibly due to a more efficient antioxidant system [5]. 

Membrane Composition and Electrostatic Interactions: The surface charge of AgNPs-cit (negative due to citrate coating) influences their electrostatic interactions with the plasma membrane. LoVo cells may present different lipid compositions or receptor profiles that enhance AgNP adhesion and uptake [6]. 

  1. Monopoli MP, Aberg C, Salvati A, Dawson KA. Biomolecular coronas provide the biological identity of nanosized materials. Nat Nanotechnol. 2012 Dec;7(12):779-86. doi: 10.1038/nnano.2012.207. PMID: 23212421.Xiao, Y., et al. (2021).  
  1. Tejeda-Muñoz N, Albrecht LV, Bui MH, De Robertis EM. Wnt canonical pathway activates macropinocytosis and lysosomal degradation of extracellular proteins. Proc Natl Acad Sci U S A. 2019 May 21;116(21):10402-10411. doi: 10.1073/pnas.1903506116. Epub 2019 May 6. PMID: 31061124; PMCID: PMC6534993. 
  1. Miethling-Graff R, Rumpker R, Richter M, Verano-Braga T, Kjeldsen F, Brewer J, Hoyland J, Rubahn HG, Erdmann H. Exposure to silver nanoparticles induces size- and dose-dependent oxidative stress and cytotoxicity in human colon carcinoma cells. Toxicol In Vitro. 2014 Oct;28(7):1280-9. doi: 10.1016/j.tiv.2014.06.005. Epub 2014 Jul 2. PMID: 24997297. 
  1. Mahgoub E, Taneera J, Sulaiman N, Saber-Ayad M. The role of autophagy in colorectal cancer: Impact on pathogenesis and implications in therapy. Front Med (Lausanne). 2022 Sep 7;9:959348. doi: 10.3389/fmed.2022.959348. PMID: 36160153; PMCID: PMC9490268.  
  1. Cai L, Sun Y, Wang K, Guan W, Yue J, Li J, Wang R, Wang L. The Better Survival of MSI Subtype Is Associated With the Oxidative Stress Related Pathways in Gastric Cancer. Front Oncol. 2020 Jul 28;10:1269. doi: 10.3389/fonc.2020.01269. PMID: 32850385; PMCID: PMC7399340. 
  1. Abruzzo A, Zuccheri G, Belluti F, Provenzano S, Verardi L, Bigucci F, Cerchiara T, Luppi B, Calonghi N. Chitosan nanoparticles for lipophilic anticancer drug delivery: Development, characterization and in vitro studies on HT29 cancer cells. Colloids Surf B Biointerfaces. 2016 Sep 1;145:362-372. doi: 10.1016/j.colsurfb.2016.05.023. Epub 2016 May 10. PMID: 27214786. 

According to the referee’s comment, we have provided an expanded discussion on the mechanisms and included additional references to support our claims, beyond reference [17]. “This heightened uptake contributes to the greater intracellular accumulation of AgNPs-cit in LoVo cells, amplifying their cytotoxic effects [21]. This is consistent with studies showing that citrate-coated nanoparticles have a higher affinity for membrane receptors involved in phagocytosis and endocytosis [22]” 

  1. Is there a discussion on the potential pathways involved in AgNP-induced programmed cell death? If so, please ensure this is clearly presented. 

Reply: 

Our results suggest that citrate-coated silver nanoparticles (AgNPs-cit) selectively induce programmed cell death (PCD) in LoVo cells through a combination of oxidative stress, mitochondrial dysfunction, and apoptosis-related signaling pathways. The major pathways implicated are as follows: 

AgNPs-cit could induce reactive oxygen species (ROS) accumulation, potentially leading to oxidative damage in LoVo cells. Increased ROS levels result in mitochondrial membrane potential (MMP) disruption, leading to cytochrome c release into the cytosol, which activates the intrinsic (mitochondrial) apoptotic pathway [1]. The significant cytotoxic effects observed in LoVo cells, but not in HT-29 cells, may be attributed to the fact that MSI-H cancer cells (e.g., LoVo) often have impaired antioxidant defense mechanisms, making them more susceptible to ROS-mediated apoptosis [2]. Morphological changes observed in LoVo cells, such as membrane blebbing, cytoplasmic shrinkage, and nuclear condensation (Figure 5), are hallmarks of caspase-dependent apoptosis [3]. While autophagy is generally a cell survival mechanism, excessive autophagic activity in response to nanoparticle-induced stress can shift toward autophagy-dependent cell death. The balance between autophagy and apoptosis may be influenced by the molecular background of the cells, as MSI-H cells are known to have altered autophagic responses [4]. 

The difference in response between LoVo (MSI-H) and HT-29 (MSS) cells may also be linked to the p53 status. LoVo cells harbor a wild-type p53, which enhances their susceptibility to stress-induced apoptosis. In contrast, HT-29 cells carry a mutated TP53, which may render them more resistant to AgNP-induced apoptosis [5]. 

To ensure clarity, we have explicitly discussed these mechanisms in Section 4 (Discussion) of the revised manuscript and cited additional supporting literature as follow: The morphological changes observed in LoVo cells, prominently displayed in Figure 5, are characteristic features of caspase-dependent apoptosis [17] 

  1. de Oliveira MR. Evidence for genistein as a mitochondriotropic molecule. Mitochondrion. 2016 Jul;29:35-44. doi: 10.1016/j.mito.2016.05.005. Epub 2016 May 17. PMID: 27223841. 
  1. Miethling-Graff R, Rumpker R, Richter M, Verano-Braga T, Kjeldsen F, Brewer J, Hoyland J, Rubahn HG, Erdmann H. Exposure to silver nanoparticles induces size- and dose-dependent oxidative stress and cytotoxicity in human colon carcinoma cells. Toxicol In Vitro. 2014 Oct;28(7):1280-9. doi: 10.1016/j.tiv.2014.06.005. Epub 2014 Jul 2. PMID: 24997297. 
  1. Aoki K, Satoi S, Harada S, Uchida S, Iwasa Y, Ikenouchi J. Coordinated changes in cell membrane and cytoplasm during maturation of apoptotic bleb. Mol Biol Cell. 2020 Apr 1;31(8):833-844. doi: 10.1091/mbc.E19-12-0691. Epub 2020 Feb 12. PMID: 32049595; PMCID: PMC7185959. 
  1. Mahgoub E, Taneera J, Sulaiman N, Saber-Ayad M. The role of autophagy in colorectal cancer: Impact on pathogenesis and implications in therapy. Front Med (Lausanne). 2022 Sep 7;9:959348. doi: 10.3389/fmed.2022.959348. PMID: 36160153; PMCID: PMC9490268. 
  1. Sritharan S, Sivalingam N. Curcumin induced apoptosis is mediated through oxidative stress in mutated p53 and wild type p53 colon adenocarcinoma cell lines. J Biochem Mol Toxicol. 2021 Jan;35(1):e22616. doi: 10.1002/jbt.22616. Epub 2020 Aug 31. PMID: 32864863. 

  1. The methodology section should better explain how the authors accounted for differences in toxicity between AgNPs-cit and AgNPs-EG6OH. Providing more details would improve clarity. 

Reply: 

To systematically compare the cytotoxic effects of AgNPs-cit and AgNPs-EG6OH, we designed our methodology to account for physicochemical properties, exposure conditions, and cellular responses as follows: 

  1. Nanoparticle Synthesis and Characterization 
  • Both AgNPs-cit and AgNPs-EG6OH were synthesized following standardized protocols to ensure consistent size, shape, and colloidal stability (see Section 2.1, Synthesis and Functionalization of AgNPs). 
  • UV-Vis spectroscopy, dynamic light scattering (DLS), and ζ-potential measurements were performed to assess differences in aggregation potential and surface charge, as these factors influence cellular uptake and cytotoxicity. 
  1. Standardized Exposure Conditions for Cytotoxicity Assays 
  • To minimize variability, all cell lines (LoVo, HT-29, and primary colonocytes) were exposed to the same nanoparticle concentration (20 µg/mL) and incubation times (24 and 48 hours). This ensures that any observed differences in cytotoxicity are attributable to nanoparticle functionalization rather than differences in dose or exposure duration. 
  • Control groups (vehicle-treated cells) were included for all conditions to normalize and compare viability data across treatments. 
  1. Assessment of Cell Viability and Toxicity 
  • We used the MTT assay to quantify cell viability, ensuring that differences in metabolic activity between cell lines were accounted for by normalizing results to untreated controls. 
  • Fluorescence microscopy and confocal imaging were used to visually confirm nanoparticle internalization and assess morphological changes indicative of cytotoxicity. 
  1. Internalization Studies to Correlate Uptake with Cytotoxicity 
  • Confocal microscopy and fluorophore-conjugated phalloidin staining were performed to evaluate intracellular nanoparticle distribution and verify that AgNPs-cit and AgNPs-EG6OH were taken up at different levels in CRC cells and normal colonocytes. 
  • The presence of dark regions (fluorescence quenching) in the cytoplasm of treated cells indicated successful nanoparticle internalization, helping to correlate uptake with toxicity outcomes. 
  1. Statistical Analysis to Validate Observed Differences 
  • Statistical analyses were conducted using GraphPad Prism 8, with unpaired t-tests for two-group comparisons and one-way ANOVA with Tukey’s post hoc test for three or more groups. 
  • The results are presented as mean ± standard deviation (SD) from at least four independent experiments, ensuring statistical robustness in assessing differences between AgNPs-cit and AgNPs-EG6OH toxicity. 
  • Statistical significance was set at p < 0.05, with asterisks denoting significant differences in Figures 2–4. 

To improve clarity, we have expanded Section 2 (Materials and Methods) to explicitly describe how differences in cytotoxicity were assessed, including additional details on: 

  • Nanoparticle characterization (Section 2.1) 

Statistical validation (Section 2.6, Statistical Analysis) 

We have revised the Materials and Methods section to ensure that the methodology clearly explains how we accounted for toxicity differences between AgNPs-cit and AgNPs-EG6OH. These revisions provide greater transparency regarding exposure conditions, uptake analysis, and statistical approaches used to validate our findings.  

  1. Finally, how does this study compare with previous research on AgNP cytotoxicity in CRC cells? A discussion on this aspect would help contextualize the findings. 

Despite the considerable attention paid to the anti-cancer effects of AgNPs in recent years, research on their potential impact in cancer remains limited. Several studies have explored the cytotoxic effects of AgNPs on secondary colon cell lines and in vivo animal models, emphasizing key factors such as particle size, surface functionalization, oxidative stress generation, and selective toxicity. However, while these studies have provided valuable insights into the mechanisms of AgNP-induced cytotoxicity, their focus has largely been restricted to transformed cancer secondary cell lines, which may not fully mimic the complexity. 

Our study aims to extend current knowledge by investigating the effects of AgNP exposure with different surface coatings not only on secondary tumor cells but also on primary colonocyte cultures derived from healthy mucosal tissues (ex vivo). Assessing the response of non-tumoral cells is crucial to determining whether AgNPs exhibit selective toxicity toward cancer cells while sparing normal tissues. Furthermore, understanding how different surface functionalization influence cellular uptake and overall cytotoxicity in both tumor and normal colonocytes will provide critical insights for optimizing AgNP-based therapeutic strategies with improved efficacy and reduced off-target effects. 

In the manuscript, we have reinforced these aspects in the Introduction section: 

  • We have added “Many works have shown that the effect of coating agent plays an important role on selective cytotoxicity, identifying that the more stable the binding of the metallic surface with the coating agent, the lower its toxicity” and the relative references 7 and 8. 
  • The sentence “Furthermore, the effect of these nanoparticles on primary colonocyte cultures derived from normal mucosal tissues of patients remains largely unexplored, with the aim of assessing whether non-tumoral cells exhibit a different response to treatment with these agents” Has been replaced with “Moreover, the impact of these nanoparticles on primary colonocyte cultures derived from normal mucosal tissues remains largely unexplored. Investigating whether non-tumoral cells exhibit a distinct response to these agents is crucial for a more comprehensive understanding of their therapeutic potential and safety” 

Reviewer 2 Report

Comments and Suggestions for Authors

Marianna Barbalinardo et al. investigated the use of silver nanoparticles as a potential new treatment for colorectal cancer. It was found that modifying the surface of silver nanoparticles could help target cancer cells in vitro, resulting in more effective but fewer side effects. The authors claimed that the work could pave the way for new cancer therapies. Basically, this design and research is too simple and superficial, and the paper is too short, especially without any experiment in vivo. Especially, I think the authors overstated their conclusions at this version based on the following major points.

  1. There was no any molecular recognition molecules grafting on the surface of silver nanoparticles. How did these nanoparticles target cancer cells?
  2. From Figure 3 and 4, no significant cytotoxicity appears in any group. So, there is no obvious effective effect for cancer treatment in vitro.
  3.  There is lack of key experiment in vivo,relevant protein and gene analysis.

Comments on the Quality of English Language

English is OK

Reply to Reviewer 2’s comments

  1. There was no any molecular recognition molecules grafting on the surface of silver nanoparticles. How did these nanoparticles target cancer cells? 

Reply: 

As we have previously reported, the formation of the protein corona plays a crucial role in mediating the cellular uptake and cytotoxicity of AgNPs (see Ref 7 in the article). It is not necessary to graft specific molecular recognition molecules onto the nanoparticle surface to achieve interaction with cells. Instead, the adsorption of biomolecules present in the cell culture medium facilitates nanoparticle internalization, allowing them to exert their cytotoxic effects. However, the formation of the protein corona does not always enhance nanoparticle-cell interactions; in some cases, it can have the opposite effect. In our specific study, we hypothesize that the interaction between nanoparticles and cells, particularly in the case of Lovo cells, is promoted by the formation of the protein corona. This is supported by the observation that AgNPs-cit capable of forming a protein corona exhibit cytotoxicity, whereas AgNPs-EG6OH, which hinder protein corona formation, do not display cytotoxic effects. 

To further clarify this point, we have added the following sentence in the discussion section: “Similarly, we can infer that the uptake of AgNPs-cit in LoVo cells is promoted by the formation of the protein corona. This heightened uptake contributes to the greater intracellular accumulation of AgNPs-cit in LoVo cells, amplifying their cytotoxic effects” 

  1. From Figure 3 and 4, no significant cytotoxicity appears in any group. So, there is no obvious effective effect for cancer treatment in vitro. 

We agree with the referee’s comment that Figures 3 and 4 show no significant cytotoxicity in HT-29 cells and primary colonocytes treated with AgNPs-cit or AgNPs-EG6OH. However, Figure 2 presents the viability data of LoVo cells treated with AgNPs-cit or AgNPs-EG6OH, revealing a significant difference. CRC is highly heterogeneous, with distinct subtypes responding differently to treatments, which is why we used not only colonocytes but also HT-29 cells (MSS, BRAF-mutant) and LoVo cells (MSI-H, KRAS-mutant). This approach allowed us to identify both a cell type-specific selectivity—LoVo cells were highly sensitive to AgNPs, unlike HT-29 cells—and the importance of nanoparticle surface properties, as AgNPs-cit were cytotoxic, whereas those coated with EG6OH were not. Overall, the results highlight how nanoparticles can interact differently with colon cells, depending both on the type of cell line and their surface properties. Therefore, in our opinion, it is important to present all the results, including those in Figure 2 as well as Figures 3 and 4, because the conclusions drawn emerge from the comprehensive analysis of these data. 

  1. There is lack of key experiment in vivo, relevant protein and gene analysis. 

Reply: 

Regarding the lack of in vivo experiments and additional protein and gene analysis, we acknowledge that these aspects would further strengthen the study. However, our research primarily aims to provide an initial in vitro evaluation of the differential cytotoxicity of surface-functionalized silver nanoparticles on colorectal cancer cells and primary human colonocytes. In vivo studies, while essential for translational research, require extensive ethical approvals and resources that extend beyond the current scope of our investigation. Nevertheless, our findings offer a critical foundation for future in vivo studies aimed at validating the selective cytotoxicity observed in our in vitro models. Similarly, while additional molecular analyses of specific protein and gene expression changes could provide further mechanistic insights, our study focuses on cellular responses at the phenotypic level. The observed differences in cytotoxicity between LoVo and HT-29 cells, as well as the absence of toxicity in normal colonocytes, are in line with previous literature on nanoparticle functionalization and its effects on cancer cell viability. 

Reviewer 3 Report

Comments and Suggestions for Authors

This study investigates the cytotoxic effects of citrate-coated (AgNPs-cit) and EG6OH-coated (AgNPs-EG6OH) silver nanoparticles on colorectal cancer (CRC) cell lines and healthy colonocytes. There are some issues that need to be resolved.

  1. The authors’ characterization of Ag modification is insufficient. To ensure that citrate-coated AgNPs can be successfully functionalized with EG₆OH, could you perform additional characterization using the following methods?

FTIR analysis – Check for disappearance of citrate (-COO⁻) peak and appearance of PEG-related C-O-C stretching.

DLS and Zeta potential – Measure hydrodynamic diameter (slight increase is expected) and surface charge.

XPS analysis – Confirm presence of C-O-C bonds and reduction of citrate-related peaks.

TGA (if available) – Compare weight loss to quantify PEG ligand attachment.

  1. In the cytotoxicity experiment, the authors should add Ag nanoparticles as a control. In addition, the authors should investigate different concentrations of AgNPs.
  2. fig2 and fig3 should be merged together. These two are the same study, just with different cells.
  3. The authors state in the Discussion that "This study also highlights the importance of understanding how specific genetic mutations and cellular properties affect nanoparticle endocytosis, reactive oxygen species (ROS) production, and apoptosis." Is this statement supported by data?
  4. The authors mention in the that "this study also highlights the importance of understanding how specific genetic mutations and cellular properties affect nanoparticle endocytosis, reactive oxygen species (ROS) production, and apoptosis." Is this statement supported by data? Similarly, "EG6OH coating may help to improve tumor-specific enrichment while reducing systemic side effects, making it important in optimizing selective cytotoxicity." This speculative statement without data support is unacceptable in a research paper

Reply to Reviewer 3’s comments 

This study investigates the cytotoxic effects of citrate-coated (AgNPs-cit) and EG6OH-coated (AgNPs-EG6OH) silver nanoparticles on colorectal cancer (CRC) cell lines and healthy colonocytes. There are some issues that need to be resolved. 

  1. The authors’ characterization of Ag modification is insufficient. To ensure that citrate-coated AgNPs can be successfully functionalized with EG₆OH, could you perform additional characterization using the following methods? 

FTIR analysis – Check for disappearance of citrate (-COO⁻) peak and appearance of PEG-related C-O-C stretching. 

DLS and Zeta potential – Measure hydrodynamic diameter (slight increase is expected) and surface charge. 

XPS analysis – Confirm presence of C-O-C bonds and reduction of citrate-related peaks. 

TGA (if available) – Compare weight loss to quantify PEG ligand attachment. 

Reply: 

The specific systems used in our study, namely citrate- and EG6OH -coated silver nanoparticles, have already been reported and extensively characterized in our previous works, where the complete replacement of citrate with thiol was demonstrated. For this reason, we considered it unnecessary to include all these additional characterizations. However, we appreciate the reviewer’s comment as it gives us the opportunity to provide further details. Specifically, we have added the hydrodynamic diameter values before and after functionalization, as well as the zeta potential measurements (DLS and Zera potential measurements). Both measurements further confirm the successful functionalization of the nanoparticles. In detail, in section 3.1.1 we have added: 

  • Citrate-coated silver nanoparticles (AgNPs-cit), with a hydrodynamic diameter of 19 ± 1 nm and a zeta potential of -44 ± 3 mV, were prepared and functionalized with EG6OH via ligand exchange… 
  • Additionally, a slight increase in hydrodynamic diameter (22 ± 1 nm) and a drastic decrease in surface charge (-19 ± 3 mV) were observed, as expected [7, 8]. 

  1. In the cytotoxicity experiment, the authors should add Ag nanoparticles as a control. In addition, the authors should investigate different concentrations of AgNPs. 

Reply: 

We thank the reviewer for giving us the opportunity to clarify this point, which may have been confusing. We tested the cytotoxic activity of AgNPs on both in vitro and ex vivo cells, using the vehicle solution in which the nanoparticles are dispersed as a control. 

To clarify this further, we have: 

  • Removed the word “untreated” before “control” in the main text and captions, as it could cause confusion for readers. 
  • Added the sentence, “For the control, the cells were exposed to the vehicle solution,” in the “Treatments of human cell culture” subsection of the Materials and Methods section. This sentence was already present in the “Human secondary cell culture” subsection. 

Regarding the choice of concentration, we focused on a single concentration because we have already reported the effects of different concentrations of AgNPs in a previous study. Based on those results, we selected a concentration at which citrate-coated nanoparticles exhibit toxicity in model cell systems, while EG6OH-coated nanoparticles do not. This approach allows us to specifically discriminate the effect of different nanoparticle coatings. To further clarify this point, we have added the following sentence in the main text: “This concentration of AgNPs, according to previously reported results, allows us to distinguish the role of surface coating in the interaction of AgNPs with cells, as under these conditions, AgNPs-citrate are toxic, whereas AgNPs completely coated with EG6OH do not significantly affect cell viability” 

  1. fig2 and fig3 should be merged together. These two are the same study, just with different cells. 

Reply: 

Thank you for your comment. However, we prefer to keep the figures separate, as merging them would reduce the size of the optical images, making it more difficult for the reader to distinguish the differences between them. 

  1. The authors state in the Discussion that "This study also highlights the importance of understanding how specific genetic mutations and cellular properties affect nanoparticle endocytosis, reactive oxygen species (ROS) production, and apoptosis." Is this statement supported by data? 

Reply: 

We appreciate the reviewer’s insightful comment. The statement in the Conclusion was intended to highlight a hypothesis based on existing literature rather than direct experimental evidence from our study. To clarify this point, we have revised the text as follows: 

"This study also suggests the importance of understanding how specific genetic mutations and cellular properties may influence nanoparticle endocytosis, reactive oxygen species (ROS) production, and apoptosis." 

Additionally, we have included relevant citations in the discussion to support this hypothesis. We hope this revision addresses the reviewer’s concern. 

  1. The authors mention in the that "this study also highlights the importance of understanding how specific genetic mutations and cellular properties affect nanoparticle endocytosis, reactive oxygen species (ROS) production, and apoptosis." Is this statement supported by data? Similarly, "EG6OH coating may help to improve tumor-specific enrichment while reducing systemic side effects, making it important in optimizing selective cytotoxicity." This speculative statement without data support is unacceptable in a research paper 

Reply: 

As stated in response No. 4, we have modified the sentence “this study also highlights the importance of understanding how specific genetic mutations and cellular properties affect nanoparticle endocytosis, reactive oxygen species (ROS) production, and apoptosis” 

Similarly, we have modified the statement regarding the EG6OH coating as follows: 

"EG6OH coating may contribute to improved tumor-specific enrichment while reducing systemic side effects.” Scientific research inherently involves the formulation of hypotheses that can be tested in future studies.  We hope these modifications adequately address the reviewer’s concerns.